# Personal Factors, Living Environments, and Specialized Supports: Their Role in the Self-Determination of People with Intellectual Disability

**DOI:** 10.3390/bs13070530

**Published:** 2023-06-24

**Authors:** Eva Vicente, Patricia Pérez-Curiel, Cristina Mumbardó-Adam, Verónica M. Guillén, María-Ángeles Bravo-Álvarez

**Affiliations:** 1Department of Psychology and Sociology, Universidad de Zaragoza, C./Pedro Cerbuna, 12, 50009 Zaragoza, Spain; 531111@unizar.es (P.P.-C.); marian@unizar.es (M.-Á.B.-Á.); 2Department of Cognition, Development and Educational Psychology, Faculty of Psychology, Universitat de Barcelona, 08035 Barcelona, Spain; cmumbardo@ub.edu; 3Department of Education, Universidad de Cantabria, Av./de los Castros, 52, 39005 Santander, Spain; guillenvm@unican.es

**Keywords:** self-determination, intellectual disabilities, context, supports

## Abstract

The self-determination of people with disabilities, and specifically people with intellectual disabilities (ID), is a growing issue due to its relevance in the field of inclusion and human rights. Although research has shown a significant relationship between self-determination and intelligence, other factors also contribute to its development. The purpose of this study was to understand what other variables may be influencing self-determination. Using the scores from 483 adolescents and adults with ID who completed the AUTODDIS scale, we performed inferential and regression analyses to determine the relationships between levels of self-determination, personal variables (sex, age, severity of ID), and contextual variables (living environment, specialized supports). We found that self-determination is affected by the severity of ID, and when this variable is controlled for, greater self-determination is mainly related to receiving occupational support and support for autonomy and independent living. Results also showed that, together with ID severity, occupational and psychoeducational support, as well as support for autonomy and independent living, were also predictors of the level of self-determination. In conclusion, this study confirms the importance of contextual variables in the development of self-determination in people with ID, placing the focus of intervention on social opportunities.

## 1. Introduction

In recent decades, self-determination has assumed an increasingly important role in the lives of people with intellectual disabilities (ID). As this construct has grown in significance, so has the theoretical development and research that aim to operationalize it, provide reliable instruments for evaluation and intervention, and comprehend the link between self-determination and other elements. Wehmeyer [1], a leading author on this topic, has defined self-determination as “volitional actions that enable the person to act as the main causal agent of his or her own life and to maintain or improve his or her quality of life” (p. 117). More recently, the functional model of self-determination [2] has been redefined as causal agency theory [3], which describes self-determination as “a dispositional characteristic that manifests itself when one acts as a causal agent of one’s own life” (p. 257). Becoming a “causal agent” is the key concept for understanding self-determination, which simply implies taking control of one’s own life and acting in accordance with one’s own principles and aspirations. The current theoretical framework [3] operationalizes self-determination in three key dimensions. These dimensions involve setting in motion a series of actions aimed at goal setting (volitional actions) and goal achievement (agentic actions); these actions, in turn, nurture the person’s own beliefs (action-control beliefs) in their ability to keep setting those actions in motion. The self-determination inventory [4,5] and the AUTODDIS scale [6,7] are assessment instruments designed to examine self-determination and its essential factors and components in relation to this theoretical framework.

The development, manifestation, and level of self-determination can be influenced by a variety of circumstances. Research initially focused on individual characteristics, including sex, age, or ID severity. Findings on sex remain unclear, with some studies [8,9] suggesting that there are differences between men and women, and others finding that there are none [10,11]. Regarding age, self-determination has conceptually been correlated with the process of development, with adolescence being acknowledged as an especially important phase for the growth of self-determination [12]. Adolescence is a time when individuals begin to take control of their own lives, contemplate their personal preferences and choices, and make plans and decisions that align with their wishes [13], although they are still developing and frequently need guidance from their support network (e.g., family members, teachers, friends and peers, professionals, and other close individuals) [14]. However, research has once again revealed inconsistent results. Some studies have found that self-determination increases during adolescence [12,15] or even in later life, particularly in specific dimensions of the construct [10]. In contrast, other studies have not detected these variations in adolescents [16] or found evidence of constancy when measuring self-determination in youth and adults [6].

In terms of the severity of ID, an association has traditionally been found between IQ and self-determination [17,18]. Research has suggested that pupils with higher degrees of ID tend to score substantially worse on self-determination measures than their counterparts with lower levels of ID [16,19]. However, Vicente et al. [16] reported that the level of support needs of people with ID, rather than severity in regard to intellectual functioning, was a key predictor of the degree of self-determination.

Beyond these personal factors, contextual factors concerning educational, family, and socio-occupational environments have also been explored. According to experts, the variety and opportunities of a person’s environment play an important role in empowering them to function in a self-determined manner [20,21]. Yet all too frequently, opportunities for people with disabilities are inadequate in social settings [22], at home, or in educational institutions [23].

Wehmeyer and Bolding [24,25] demonstrated that the degree of self-determination of people with ID was substantially correlated with the restrictiveness or inclusiveness of living and working environments. Álvarez-Aguado et al. [26] showed that while people living independently outside the family home exhibited more autonomy, they did not score higher in any of the other self-determination dimensions. However, in addition to the location and static characteristics of the setting, the provision of support and opportunities in such contexts could play a fundamental role in self-determination. For instance, situations that are less welcoming or normalized are more likely to limit people’s abilities to make decisions and act as the causal agent of their own lives [18]. Depending on the presence and severity of a disability, different environmental opportunities have distinct effects on aspects of self-determination [27]. In this regard, the availability of opportunities for self-determination is a substantial predictor of self-determined actions [8].

Likewise, the type of support and services a person receives in the context in which they live is also an important aspect that can influence self-determination. In a study by Álvarez-Aguado et al. [26], people who received support from their families showed higher levels of self-determination compared with those who only received professional support.

There is a clear demand for additional research on this topic, delving deeper into the specific role of support provision in self-determination and differentiating between the various types of support services as catalysts of self-determination.

The concepts of quality of life and supports have recently been combined to create the quality of life supports model (QOLSM) [28], which posits that effectively managed individual supports provided by the environment, rather than just individual functioning rehabilitation, are what lead to improvements in personal satisfaction and quality of life (QOL) [29]. This model, which is being widely used internationally to provide support, reform organizations, and modify systems, proposes four clear components: (1) core values (including an emphasis on self-determination, equity, and inclusion); (2) QOL domains (with self-determination constituting one of those domains according to one of the most internationally recognized theoretical models [30]; (3) support systems (which provide the framework for improving individual or family functioning and well-being); and (4) facilitating conditions (which are contextual factors that help to apply the model, ranging from community involvement or a sense of permanence to the availability, accessibility, and personalization of supports). According to Thompson et al. [31], people with ID often encounter an imbalance between their individual competencies and the demands of the environment, leading to a variety of support needs that diverge in type and intensity. Support can be defined as resources and strategies that enrich human functioning by contributing to autonomous participation in society and potentially enhancing well-being [32]. Systems of support [33] have been defined as a broad range of resources and strategies that prevent or mitigate a disability or its effects and that promote the development, education, and well-being of people with ID or their families. These systems typically include four elements: (1) choice and personal autonomy (i.e., having opportunities to choose and exercise self-determination on an equal basis with people without disabilities); (2) inclusive educational, living, and work environments; (3) generic supports (which are widely available to the general population, including natural supports, technology, prosthetics, etc.); and (4) specialized supports (i.e., professional interventions, strategies, and therapies).

The aim of this study is to inquire into the role of personal and contextual factors in the self-determination of people with ID. Specifically, we analyze the role of each of the contextual factors (environment in which the person lives and the variety of supports services received) while adjusting for the potential impact of the degree of ID severity as a covariate. This analysis builds on prior lines of research that have drawn a link between the severity of ID and self-determination. It also aligns with the current psychosocial approach to ID, which emphasizes not only the importance but also the relevance of contextual factors and supports.

## 2. Materials and Methods

### 2.1. Participants and Settings

A total of 483 people with ID aged between 10 and 40 were recruited using an incidental sampling method (*M* = 26.28; *SD* = 8.28). The majority (60.7%) of these participants were male, had mild or moderate ID severity as measured by adaptive behavior (*n* = 385; 79.7%), and resided in the family home (62.5%). Nearly 80% of the 170 professionals who worked with them in Spanish organizations and institutions that care for individuals with disabilities were women. The assessments of the participants with IDs were completed by professionals. Each of them assessed between 1 and 37 people whom they had known for at least 4 months and whom they had opportunities to observe and engage with regularly in various settings. In all, 83.8% of the professionals had an average frequency of daily or almost daily contact with the participants.

Table 1 and Table 2 provide detailed information on the characteristics of the sample, showing the personal and contextual factors assessed, including information on whether the participants received specialized support services.

### 2.2. Instruments

#### 2.2.1. Self-Determination

The AUTODDIS scale [7,11] was used to assess self-determination. According to the most recent theoretical model [3], the scale has 47 items and is divided into six subscales: autonomy, self-initiation, self-direction, self-regulation, self-realization, and empowerment. It also provides a global self-determination score. All items must be answered on a four-point Likert scale based on level of agreement (i.e., strongly disagree, disagree, agree, and strongly agree) by a respondent who knows the person with ID well (for at least 4 months). This scale is designed to assess people with ID between the ages of 11 and 40. The scale was created after an exhaustive elaboration procedure that included (1) a Delphi study [34], (2) a pilot study [35], and (3) an in-depth analysis of the evidence of reliability and validity [7,11]. The final version of the scale exhibits concurrent validity and Cronbach’s alpha values close to or above 0.95 [7]. Research has also validated the internal organization of this instrument as well as its equivalence and measurement invariance in adolescents and adults [11].

#### 2.2.2. Sociodemographic Data

Via an initial questionnaire, participants with ID provided sociodemographic information, including details about their sex, age, and ID severity level. Professionals who completed the scale were also required to categorize the participants’ degree of ID severity by assigning them to one of four categories (mild, moderate, severe, or profound) based on the most recent DSM-5 [36] at the time the study was conducted. They were instructed to use readily available reports that contained data from standardized tests or other clinical judgments.

This questionnaire also collected contextual information about the home setting (i.e., family home, residential setting, independent living with and without supports, or a combination of several settings) and the types of resources or support services received (i.e., psychoeducational support, speech therapy, occupational support, autonomy and independent living skills, physiotherapy, or leisure services).

### 2.3. Procedure

The data used in this study come from a research project developed for the validation of the AUTODDIS scale (project PSI2016-75826-P, AEI/FEDER, EU). This project received the approval of the Ethics Committee of the Autonomous Community of Aragon (CEICA), ensuring ethical standards in its procedures.

A total of 33 organizations and institutions working with individuals with ID from 11 of Spain’s 17 autonomous communities participated in the initiative. For the purpose of this paper, we used data from 28 of them. Each organization designated a contact person who was tasked with selecting participants according to two criteria: (1) persons with ID; and (2) persons aged between 11 and 40 years. No exclusion criteria were established. The contact person was also responsible for notifying the individuals involved (participants being assessed, their families, and/or legal guardians) and for obtaining all the necessary informed consents. Similarly, the contact person selected and coordinated professionals to conduct the assessments. These assessors had to know the participants well (for at least 4 months) and be familiar with the construct of self-determination. Assessors could begin data collection only when participants (or their legal guardians) had signed their informed consent. To guarantee the confidentiality and anonymity of the data, identification codes were used to replace the first and last names of participants and assessors. Assessors had the option to record the information either online or on paper.

### 2.4. Data Analysis

Different data analyses were conducted to examine the impact of contextual factors (i.e., place of residence and various types of supports) and personal factors (i.e., severity of ID, sex, and age) on the degree of self-determination. To begin with, *t* tests and analysis of variance models (ANOVA) were created to determine whether there were any significant variations in the overall AUTODDIS scale score based on each factor. Then, in order to eliminate any potential influence from contextual factors, analysis of covariance (ANCOVA) models were developed to find any significant variations in the AUTODDIS scale’s overall score. These analyses included the individuals’ degree of ID as a covariate. The eta partial squared (η^2^) method was used to determine the effect size. Values below 0.01, above 0.06, and above 0.14 were regarded as small, moderate, and large, respectively. Third, by means of hierarchical regression analysis, the predictive ability of personal and contextual characteristics that were determined to be significant in earlier analyses was examined in relation to the overall self-determination score. The tolerance index, which must be higher than 0.05, and the variance inflation factor (VIF), which must be less than 2, were surveyed prior to conducting the regression to determine the multicollinearity between the predictors [37].

The normality, linearity, and homoscedasticity of the dependent variable residuals were explored by analyzing the shape of the residuals’ distribution and comparing the residuals’ scatterplot with the theoretical values [37]. The independence of errors was investigated via the Durbin–Watson test (which must range between 1.5 and 2.2), and the presence of outliers in the solution was detected by analyzing the standardized residuals.

## 3. Results

### 3.1. Personal and Contextual Factors Affecting Self-Determination in People with Intellectual Disabilities

When comparing the average general score of self-determination with the different personal factors, we found no significant differences (Table 3), except for the severity of ID (*p* < 0.001). The comparative post hoc contrast shows that participants with mild ID presented significantly higher means in self-determination than their counterparts with moderate and severe/profound ID. Similarly, participants with moderate ID also exhibited significantly higher means of self-determination than participants with severe/profound ID.

Significant differences were identified for all contextual factors (*p* < 0.05). First, the place of residence played a significant role in the level of self-determination (Table 4). Participants who were living in a residential setting had significantly lower means of self-determination than participants who were living in the family home (*p* < 0.001), living independently (*p* < 0.001), or living in the family home combined with other settings (*p* = 0.009). Additionally, participants who were living independently showed significantly higher levels of self-determination than those who were residing in the family home (*p* = 0.034). However, these levels were not significantly higher when compared with participants who combined living in the family home with other settings.

In terms of type of support, the data indicate two different patterns (Table 4), depending on whether the supports were more clinical or more psychosocial. Participants who were receiving speech therapy, psychoeducational support, and physiotherapy had significantly lower levels of self-determination than their counterparts who were not receiving these types of supports. In contrast, participants benefiting from occupational support, autonomy and independent living skills support, or access to leisure services exhibited a significantly higher level of self-determination when compared with participants not in receipt of these supports.

These analyses inform current psychosocial models of ID, which highlight the importance of contextual factors and supports. However, given the role that the severity of ID plays in self-determination, it was considered appropriate to perform an analysis of covariance. The aim of this step was to once again analyze the role of each of the contextual factors, but this time considering the level of ID severity as a covariate (Table 5).

The results indicate that when ID severity is controlled for as a covariate, the place in which the person resides, occupational support, autonomy, and independent living-skills support have a significant and moderate effect on the level of self-determination of people with ID. The other support services (e.g., speech therapy and leisure services) ceased to have significant effects when ID severity was incorporated as a covariate. Psychoeducational and physiotherapy supports showed significant results, albeit with a very tiny effect size.

### 3.2. Personal and Contextual Factors Predicting the Overall Self-Determination Score in People with Intellectual Disabilities

In order to determine the weight of the predictor variables (personal and contextual variables) in the overall self-determination score, a regression analysis was performed. The resulting model complied with the assumptions of linearity, normally distributed residuals, and a lack of multicollinearity. VIF indexes were lower than 2, ranging from 1.012 to 1.134, and tolerance indexes were higher than 0.05, ranging from 0.882 to 0.988. A result of 1.56 indicated that the Durbin–Watson test’s requirement for independence of errors was satisfied. The results of the linear regression analysis revealed that support for autonomy and independent living, psychoeducational support, occupational supports, and ID severity all strongly predicted the overall scores for self-determination.

Table 6 displays the results of the hierarchical regression analysis with the self-determination total score as the dependent variable, ID severity as the independent variable (first step), and psychoeducational, occupational, autonomy, and independent living supports as the independent factors (second step). First, it was discovered that the ID severity personal factor had a substantial impact on predicting self-determination, as it explained 38.8% of the variance of the self-determination total score (*R*^2^ = 0.388; *F*(1,480) = 304.272, *p* < 0.001, β = −0.623; *p* < 0.001). Second, when contextual factors were taken into account, the variance explained rose to 45.1% (*R*^2^ = 0.451; *F*(6,75) = 65.065, *p* < 0.001), and psychoeducational, occupational, autonomy, and independent living supports all predicted self-determination. The place of residence and physiotherapy support had no discernible effect on self-determination.

## 4. Discussion

The importance of self-determination for the dignity of people with disabilities was recognized with the adoption and enactment of the United Nations Convention on the Rights of Persons with Disabilities (CRPD; [38,39]). Article 12 of the CRPD emphasizes the need to respect the personal autonomy of persons with disabilities, consider their values and beliefs, and support them in the realization of their personal choices [13].

However, the risk of denying this right to people with ID still exists [40]. An example of this risk is to associate self-determination with independence and self-sufficiency [41], erroneously assuming that showing self-determination implies doing everything by oneself [1]. According to Wehmeyer, “the capacity to perform specific behaviors (e.g., independently) is secondary in importance in being self-determined to whether one acts volitionally and makes things happen in one’s life” [1] (p. 115).

In addition to regarding it as a right, research shows a connection between promoting self-determination and better academic performance as well as the accomplishment of more favorable outcomes in adulthood (e.g., inclusion in the community and work, independent living) [3,39,42,43].

With this in mind, there is therefore a clear need for continued research in this area, to identify factors that encourage and promote self-determination. Personal and contextual factors should be considered together, as one type mutually reinforces the other. It is in continuous synergy that favorable scenarios are generated for the promotion, development, acquisition, and maintenance of self-determined behaviors.

The findings of this study indicate that individual characteristics such as sex or age have no effect on one’s capacity for self-determination.

Previous research into the influence of sex on self-determination has been contradictory [9]. The dimensions and components of the construct should be explored in greater depth in order to find differences. For instance, it has been suggested that women are less likely than their male peers to describe themselves as agentic and competent in leadership [44]. Additionally, the sociocultural environment should be considered when researching this aspect [45], and the results should be evaluated in relation to the context [33,46,47]. Despite the current absence of conclusive findings, future research should nonetheless deepen its focus on the role of sex in the expression of self-determination and its dimensions in order to propose interventions from this perspective. The same can be said of age. Although our study did not find age to be a key factor in the self-determination of participants, it may be appropriate to continue investigating whether self-determination needs vary depending on where individuals are in their lifespan.

Our study confirmed ID severity as one of the primary determinants of self-determination. These findings were consistent with earlier research [17,21], which suggested that participants with milder ID showed better levels of self-determination than their counterparts with more severe ID. However, other studies [34] have indicated that this relationship is complex (and even nonsignificant) when other factors such as social skills or the level of support needed are taken into account. It is for this reason that we sought to explore the role of other contextual factors connected to the provision of support. We indeed found that certain contextual elements also have an impact on self-determination in addition to the personal factor of ID severity.

According to Abery and Stancliffe’s ecological model of self-determination [48], it is assumed that nothing occurs in isolation but in a social context as part of a reciprocal relationship between people with disabilities and their environment. This idea is in line with the current definition of ID, which indicates that additional factors (such as the community environment typical of the individual’s peers and culture) must be considered when assessing ID [33].

The expression of self-determination is influenced by the interaction between personal characteristics and environmental conditions. Therefore, and in line with Davy [49], the autonomy of the self is constituted in and through relationships: the environment influences individuals, and individuals interact with their environment as they learn to become more self-determined.

As a result, our study included participants’ place of residence as a contextual variable, and the findings indicate that more inclusive environments encourage the expression of self-determination. Although this variable did not predict levels of self-determination, there were significant differences between participants based on where they lived, with people in residential settings scoring lower on this metric than people living in any of the other types of settings, even after accounting for the effect of ID severity in the analysis. These findings are consistent with earlier research [50], which demonstrated that living independently (with or without supports) and in inclusive surroundings gave individuals opportunities to utilize all of the abilities associated with self-determination more fully. Similarly, Cudré-Mauroux [51] argued that individuals with ID are better able to achieve emancipation if they have relationships where they can expect to receive support when they need it.

In addition to the physical environment in which a person resides, the key to progress in personal satisfaction and QOL, according to the current QOLSM [28], lies in the effective management of the support systems available in that environment. For this reason, in this paper we have also investigated the role that specialized supports for people with ID may play in self-determination.

Significant differences were found in the levels of self-determination between participants who did and did not receive speech therapy, physiotherapy, or psychoeducational support, with greater self-determination in those who did not receive this type of support. Physiotherapy and psychoeducational support were significant when the effect of severity was controlled for, although both had a small effect size. In contrast, speech therapy ceased to have an effect on self-determination when ID severity was incorporated as a covariate. This result may be associated with the fact that self-determination skills are related to language and other cognitive and motor skills [52], with communication skills considered a prerequisite for self-determination [53]. Having a means to communicate with and be understood by others (especially for individuals with high support needs) is vital for expression and the development of self-determination. Thus, individuals who have greater language proficiency or a functional communication system—and therefore do not require speech therapy support—tend to have higher levels of self-determination.

In contrast, participants receiving occupational support or support for autonomy and independent living had higher self-determination scores, even when ID severity was controlled for and with medium effect sizes. Both types of support are aimed at promoting skills such as autonomy, daily living skills, or decision-making, and they are therefore key to promoting self-determination in people with ID. Consequently, professionals working with people with ID should be mindful of the importance of this type of support. Likewise, research should delve deeper into the specific characteristics that this type of support should have in order to contribute to the development of self-determination and its different components. For example, there is empirical evidence that the use of ICTs (Information and Communications Technologies) can help people with disabilities improve specific skills [10,54], which can also contribute to self-determination.

Finally, our analyses concurred with previous research suggesting that ID severity has a key role in explaining the level of self-determination. Importantly, they showed that, together with ID severity, occupational support, psychoeducational support, and support for autonomy and independent living were also predictors of the level of self-determination.

As with all research, our study is not without its limitations. These include the incidental nature of the sample, which limits the generalizability of our findings. Furthermore, although every effort was made to include the different types of support available to participants, other contextual factors not considered here may also influence levels of self-determination. We chose to focus on specialized supports but knowing more about a person’s informal support networks [55,56] as well as other contextual factors, including sociocultural climate, school environment, and family values or expectations, could be equally relevant and significantly influence the promotion of self-determination [57,58]. Likewise, the procedure used to assess personal and contextual variables—third-party information with variables using dichotomous YES/NO responses or categories—may be a limitation. All that is known is whether the person receives the support, with no information about the characteristics and needs of that support. Additionally, the level of ID is based on an estimate given by professionals using readily available reports or their clinical judgments, but there is no control over the evaluation process of this variable. Finally, although the selected statistical methods provide us with relevant information, by taking into account the independent effect of each variable on self-determination, it would be interesting to broaden the research by including other statistical methods (such as, performing mediation and moderation analyses or longitudinal studies that allow us to delve into the results).

In spite of these limitations, our results add to the scant research that specifically investigates specialized support services and their influence on self-determination, which is vital when planning supports adjusted to an individual’s needs. In this way, our findings mark out a useful starting point that will certainly open up possibilities for future avenues of inquiry.

## 5. Conclusions

In recent years, many countries, particularly those that have ratified the CRPD, have implemented a range of social policies to improve the support and participation of people with ID. Yet the fact remains that these individuals still have limited opportunities to make decisions, express their preferences, and ultimately exercise their right to self-determination.

Research has shown that both personal and contextual factors can predict levels of self-determination in people with ID. Although ID severity was the strongest predictor of self-determination in this research, given that the types of support the person receives will be closely linked to the degree of severity, it is important to consider other contextual variables (some of which have been analyzed here) that may also play a role. Another relevant aspect to consider is the construct of support needs [16], which differs from classification factors traditionally linked to intellectual functioning or even adaptive behavior [33]. Talking about people with greater support needs (rather than with greater severity of ID) is a more comprehensive approach to understanding ID [33], reflecting the person’s potential and providing a framework for implementing systems of support. From this approach, it is logical to assume that contextual support can also be predictors of self-determination for people with ID. It shows that all individuals can enhance self-determination when appropriate supports and opportunities to engage in self-determined actions are provided, especially when these supports are aligned with the needs of the individual. This matching of support to needs should be the focus of future research. Self-determined people make things happen in their lives, and contexts can support the development of skills linked to self-determination. These skills are developed over time through participation in opportunities to apply them, through direct instruction, and through appropriate support.

From this perspective, it is vital to achieve a better understanding of self-determination and the factors that may be contributing to its development in order to be able to provide tailored support.

## Figures and Tables

**Table 1 behavsci-13-00530-t001:** Personal Factors. Frequency and Percentage of Participants.

Variable	Indicators	Frequency (%)
Sex	Male	293 (60.7)
	Female	190 (39.3)
Age range	10–20	100 (20.7)
	21–30	197 (40.8)
	31–40	186 (38.5)
ID severity (estimation based on adaptive behavior)	Mild	180 (37.3)
	Moderate	205 (42.4)
	Severe/Profound	98 (20.2)

**Table 2 behavsci-13-00530-t002:** Contextual Factors. Frequency and Percentage of Participants.

Variable	Indicators	Frequency (%)
Place of residence	Family home	302 (62.5)
	Residential setting	136 (28.2)
	Independent living (with or without supports)	34 (7.0)
	Family home + other settings	11 (2.3)
Psychoeducational support	YES	233 (48.2)
	NO	249 (51.6)
	Missing	1 (0.2)
Speech therapy support	YES	106 (21.9)
	NO	376 (77.8)
	Missing	1 (0.2)
Occupational support	YES	247 (51.1)
	NO	235 (48.7)
	Missing	1 (0.2)
Autonomy and independent living skills supports	YES	381 (78.9)
	NO	101 (20.9)
	Missing	1 (0.2)
Physiotherapy support	YES	105 (21.7)
	NO	377 (78.1)
	Missing	1 (0.2)
Leisure services	YES	105 (21.7)
	NO	377 (78.1)
	Missing	1 (0.2)

**Table 3 behavsci-13-00530-t003:** Means and Standard Deviations for the Self-Determination Score by Personal Factor.

Variable		Self-Determination		
		*M*(*SD*)	*F*/*t*	*p*
Sex	Male	112.18 (29.23)	−1.705	0.089
	Female	116.70 (27.11)		
	11–20	111.89 (26.22)	1.876	0.154
Age range	21–30	116.97 (28.41)		
	31–40	111.87 (29.54)		
ID severity	Mild	131.07 (19.61)	167.21	0.000
	Moderate	114.78 (21.67)		
	Severe/Profound	80.82 (26.02)		

**Table 4 behavsci-13-00530-t004:** Means and Standard Deviations for the Self-Determination Score by Contextual Factor.

Variable		Self-Determination		
		*M* (*SD*)	*F*/*t*	*p*
Place of residence	Family home	116.68 (25.42)	13.98	0.000
	Residential setting	102.59 (32.08)		
	Independent living (with or without supports)	130.15 (22.94)		
	Family home + other settings	129.72 (28.47)		
Psychoeducational support	NO	116.68 (28.94)	2.17	0.030
	YES	111.06 (27.79)		
Speech therapy support	NO	116.75 (27.64)	4.11	0.000
	YES	104.07 (29.43)		
Occupational support	NO	111.05 (30.52)	−2.19	0.029
	YES	116.73 (26.21)		
Autonomy and independent living skills support	NO	103.57 (34.30)	−4.19	0.000
	YES	116.71 (26.12)		
Physiotherapy support	NO	117.48 (26.38)	5.27	0.000
	YES	101.34 (32.19)		
Leisure services	NO	109.20 (32.55	−2.57	0.011
	YES	116.24 (26.06)		

**Table 5 behavsci-13-00530-t005:** Results of the ANCOVA Tests.

	Variables	*F*	*p*	η^2^
ANCOVA 1	ID severity	306.720	0.000	0.391
Adjusted *R*^2^ = 0.435	Place of residence	14.773	0.000	0.085
ANCOVA 2	ID severity	307.972	0.000	0.391
Adjusted *R*^2^ = 0.395	Psychoeducational support	7.382	0.007	0.015
ANCOVA 3 Adjusted *R*^2^ = 0.389	ID severity	281.335	0.000	0.370
	Speech therapy support	2.757	0.097	0.006
ANCOVA 4 Adjusted *R*^2^ = 0.416	ID severity	336.365	0.000	0.413
	Occupational support	25.044	0.000	0.050
ANCOVA 5 Adjusted *R*^2^ = 0.421	ID severity	321.809	0.000	0.402
	Autonomy and independent living skills supports	29.048	0.000	0.057
ANCOVA 6 Adjusted *R*^2^ = 0.391	ID severity	267.928	0.000	0.359
	Physiotherapy support	4.602	0.032	0.010
ANCOVA 7 Adjusted *R*^2^ = 0.389	ID severity	293.484	0.000	0.383
	Leisure services	3.181	0.075	0.007

**Table 6 behavsci-13-00530-t006:** Results of the Hierarchical Regression Analysis.

	*R* ^2^	Δ*R*^2^	Δ*F*	β	*sr* ^2^
Model 1	0.388	0.387	304.27 **		
ID severity				−0.623 **	−0.623
Model 2	0.451	0.444	65.065 **		
ID severity				−0.623 **	−0.629
Psychoeducational support				−0.097 *	−0.126
Occupational support				0.124 **	0.157
Physiotherapy support				−0.037	−0.047
Autonomy and independent living skills supports				0.157 **	0.198
Place of residence				−0.010	−0.014

Note: ** *p* < 0.001, * *p* < 0.01.

## Data Availability

Data are available upon request to the corresponding author.

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
