# Peer review of "Personal Factors, Living Environments, and Specialized Supports: Their Role in the Self-Determination of People with Intellectual Disability"

_behavsci, 2023, doi:10.3390/bs13070530_

Round 1

Reviewer 1 Report

There is a great need for research such as this one, in order to highlight the effect that environmental support has on the development of functional abilities in persons with intellectual disability. In this case, the focus is on self-determination, an aspect which, as the authors themselves say, is very important. It is indeed refreshing to see quantitative research that does not only focus on impairment but takes environmental factors into account. 

The article is well written and the research clearly produced. The research method used is explained clearly. The discussion is in depth and it refers to the main literature on self-determination and persons with intellectual disability as well as relevant studies. 

However, the argument of the importance of environmental factors and their impact on the development of self-determination skills should be more strongly presented. In this regard, the following changes are recommended: 

- in the abstract, the sentence 'Although research has shown .... to its development', places too much emphasis on the effect of intelligence. Saying that research shows 'a significant relationship between self-determination and intelligence' and then only suggesting 'that other factors contribute to its development', implies that these factors are not as important as intelligence. In fact, this research (among others) shows that this is not the case. Additionally, there are various aspects to intelligence (including, for example, emotional intelligence), which are not necessarily directly correlated with 'academic' intelligence. This sentence should therefore be reworded so that intelligence and environmental factors are given equal importance. 

- Likewise, in the conclusion the language used seems too hesitant at times. In the sentence starting 'From this approach', the authors can write 'contextual supports, 'It suggests' can be changed to 'It shows'. 

- Additionally, in the discussion the authors may consider engaging with the definition of intellectual disability provided by the AAIDD  https://www.aaidd.org/intellectual-disability/definition . The research presented in this article speaks to the importance of the 'additional considerations' outlined by AAIDD. 

Author Response

However, the argument of the importance of environmental factors and their impact on the development of self-determination skills should be more strongly presented. In this regard, the following changes are recommended: 

- in the abstract, the sentence 'Although research has shown .... to its development', places too much emphasis on the effect of intelligence. Saying that research shows 'a significant relationship between self-determination and intelligence' and then only suggesting 'that other factors contribute to its development', implies that these factors are not as important as intelligence. In fact, this research (among others) shows that this is not the case. Additionally, there are various aspects to intelligence (including, for example, emotional intelligence), which are not necessarily directly correlated with 'academic' intelligence. This sentence should therefore be reworded so that intelligence and environmental factors are given equal importance. 

Thank you for your suggestion. The sentence has been reworded.

Although research has shown a significant relationship between self-determination and intelligence, other factors also contribute to its development.

- Likewise, in the conclusion the language used seems too hesitant at times. In the sentence starting 'From this approach', the authors can write 'contextual supports, 'It suggests' can be changed to 'It shows'. 

Thank you for your comment. Changes have been made in this paragraph.

- Additionally, in the discussion the authors may consider engaging with the definition of intellectual disability provided by the AAIDD  https://www.aaidd.org/intellectual-disability/definition . The research presented in this article speaks to the importance of the 'additional considerations' outlined by AAIDD. 

Thank you for your comment. Changes have been made in this section.

According to Abery and Stancliffe’s ecological model of self-determination [48], it is assumed that nothing occurs in isolation, but in a social context, as part of a reciprocal relationship between people with disabilities and their environment. This idea is in line with the current definition of ID, which indicates that additional factors (such as the community environment typical of the individual’s peers and culture) must be taken into account when assessing ID [33].

Reviewer 2 Report

Line 388 - ICTs need to be defined.

Table 6 - "Living" should be capitalized to maintain consistency 

Good background analysis, good number of participants from broad geographical areas within Spain, appropriate variables examined, very comprehensive and accurate exploration of limitations. Discussion and conclusions are relevant, interesting, and appropriate. Definitely adds to the existing literature and inspires further research to fill in gaps.

Author Response

Good background analysis, good number of participants from broad geographical areas within Spain, appropriate variables examined, very comprehensive and accurate exploration of limitations. Discussion and conclusions are relevant, interesting, and appropriate. Definitely adds to the existing literature and inspires further research to fill in gaps.

Line 388 - ICTs need to be defined.

Thank you for your suggestion. The acronym’s meaning has been provided.

Table 6 - "Living" should be capitalized to maintain consistency 

Thank you for your comment. In this case "autonomy and independent living skills supports" is a whole concept. Due to table format does not fit on the same line.

Reviewer 3 Report

It was my pleasure to review the manuscript “Personal Factors, Living Environments, and Specialized Sup-2 ports: Their Role in the Self-Determination of People with Intellectual Disability”. This article aims to determine the relationships between levels of self-determination, personal variables (sex, age, severity of ID), and contextual variables (living environment, specialized supports).

The abstract is brief and explains the reason for the paper well.

 The introduction is well-written and explains the need for research into the role of personal and contextual factors in the self-determination of people with ID.

Material and methods

This section is well written. It was not clear how ID severity was determined and how the reliability of ID severity was assessed by a variety of professionals who participated in the assessment.

Results

Personal and contextual factors are well described.

The discussion is well written. It provides a good summary of all findings and goes over factors that promote self-determination.

The conclusion is concise and explanatory.  

Limitations of the study addressed. It would be beneficial to discuss the limits of the statistical methods selected. 

Author Response

It was my pleasure to review the manuscript “Personal Factors, Living Environments, and Specialized Sup-2 ports: Their Role in the Self-Determination of People with Intellectual Disability”. This article aims to determine the relationships between levels of self-determination, personal variables (sex, age, severity of ID), and contextual variables (living environment, specialized supports).

The abstract is brief and explains the reason for the paper well.

The introduction is well-written and explains the need for research into the role of personal and contextual factors in the self-determination of people with ID.

Results

Personal and contextual factors are well described.

The discussion is well written. It provides a good summary of all findings and goes over factors that promote self-determination.

The conclusion is concise and explanatory.  

Material and methods

This section is well written. It was not clear how ID severity was determined and how the reliability of ID severity was assessed by a variety of professionals who participated in the assessment.

Thank you for your comments and suggestions. This issue has been incorporated as a limitation of the study

Likewise, the procedure used to assess personal and contextual variables—third-party information with variables using dichotomous YES/NO responses or categories—may be a limitation. All that is known is whether the person receives the support, with no information about the characteristics and needs of that support. And, the level of ID is based on an estimate given by professionals using readily available reports or their clinical judgments, but there is no control in the evaluation process of this variable

Limitations of the study addressed. It would be beneficial to discuss the limits of the statistical methods selected. 

Thank you for your comments and suggestions. Some changes have been introduced in this section.

Finally, although the selected statistical methods provide us with relevant information, by taking into account the independent effect of each variable on self-determination, it would be interesting to broaden the research by including other statistical methods (such as, performing mediation and moderation analyzes or longitudinal studies that allow us to delve into the results).
